# Use of Sewage Sludge Ash in the Production of Innovative Bricks—An Example of a Circular Economy

Andelina Bubalo [1,*], Drazen Vouk [1], Nina Stirmer [2] and Karlo Nad [3]

1 Water Research Department, Faculty of Civil Engineering, University of Zagreb, 10000 Zagreb, Croatia; drazen.vouk@grad.unizg.hr
2 Department of Materials, Faculty of Civil Engineering, University of Zagreb, 10000 Zagreb, Croatia; nina.stirmer@grad.unizg.hr
3 Division of Experimental Physics, Ruđer Bošković Institute, 10000 Zagreb, Croatia; karlo.nad@irb.hr
* Correspondence: andelina.bubalo@grad.unizg.hr

**Abstract:** In this paper the properties of clay bricks with 5 wt%, 10 wt%, and 20 wt% sewage sludge ash (SSA) were studied and compared with the properties of control bricks made of 100% clay. Sewage sludge (SS) was collected at two wastewater treatment plants (WWTPs) in Croatia—WWTP Zagreb and WWTP Karlovac—and incinerated at a temperature of 900 °C The bricks were produced on a laboratory scale. A total of seven types of bricks were produced—control bricks and six types of bricks as combinations of different wt% of SSA generated from SS that was collected at two different WWTPs. The physical and mechanical properties of produced bricks were tested. Compressive strengths of bricks with 5 wt% SSA (54.0–54.5 N/mm$^2$) and 10 wt% SSA (50.2–51.0 N/mm$^2$) were higher compared to the control bricks (50.4 N/mm$^2$), while bricks with 20 wt% SSA (37.0–43.9) N/mm$^2$) had noticeably lower compressive strenght. The coefficient of saturation was lower for bricks with SSA compared to control bricks. The initial absorption values were more pronounced for SSA fractions of 20 wt%.

**Keywords:** sewage sludge; SSA; bricks; brick production; recycling; circular economy

## 1. Introduction

The production of bricks from clay has a long tradition in the world and its growth is particularly pronounced in developing countries. Globally, 1500 billion bricks are produced annually, 87% of which are produced in Asia [1]. The trend of urbanization and industrial expansion is leading to an increase in demand for bricks, although brick production in some countries has shown a negative trend in the last decade [1,2]. Brick is one of the most commonly used building materials. Although the possibilities of using brick are wide, it is still mainly used for the construction of residential buildings [3]. Clay brick provides the properties of a good thermal insulator, fulfills the capacity of heat storage, and is characterized by the ability to self-regulate moisture as a result of high resistance to water vapor diffusion and by good sound insulation, high fire resistance, and long service life [3,4]. On the other hand, brick production is characterized by low energy efficiency [5,6]. Calkins reported that brick production requires 150% to 400% more energy than concrete block production [6]. Brick production contributes to greenhouse gas (GHG) and black carbon (BC) emissions, which have a significant impact on human health and climate change. In addition, the brick production sector is characterized by intensive exploitation of natural resources—clay deposits. In 2014, for example, approximately 10 million tons of common clay were mined in the United States alone [7]. In order to minimize the impacts associated with natural resource exploitation, reduce production costs, and establish a higher level of environmental sustainability, nearly 50% of global manufacturers incorporate some form of waste into brick production [8]. Many researchers have also analyzed the possibility of and justification for using various waste materials in brick production, including for

example fly ash and kiln-bottom ash [9,10], coal-producer fly ash [11], mine waste [12], cigarette butts [13], and rice-husk ash [14]. The possibility of sewage sludge (SS) [15] as a substitute raw material in brick production and the possibility of mixing SS with other waste materials (e.g., fly ash, ash from the bottom of the kiln, agricultural waste, forest waste, and waste glass) has been studied since the 1980s. An adequate solution for the disposal of SS from wastewater treatment plants (WWTPs) is a global problem, and from the analysis of practices and experiences in the EU and the world in the field of SS management, no unique approach to the application of technologies for the treatment and use or disposal of SS can be identified. Wastewater treatment plants in European countries generate about 10 million tons of SS as dry matter per annum. Only thermal treatment allows this amount to be reduced by 70% by weight and 90% by volume [16]. In recent years, thermal treatment of SS by incineration or co-incineration has been increasingly promoted and this method is most widespread in Switzerland, Netherlands, Germany, Austria, and Belgium [17]. Worldwide, mainly in the USA, the EU, and Japan, about 1.7 million tons of incinerated sewage-sludge ash (SSA) are generated [18,19]. SSA is mostly disposed of in controlled landfills, although there are numerous examples of positive practices in the use of SSA in construction [20–35]. It is obvious that the developed countries in the world are positively oriented to a circular economy and looking for technologies to make use of SS through its complete digestion by thermal treatment (with maximum energy recovery) and with the use of SSA (agricultural and non-agricultural land, construction, etc.). The incorporation of SSA into bricks fits well into the above example of a circular economy and helps to reduce the use of raw clay from nature, thus conserving natural resources. The production of bricks with SSA involves the usual steps of mixing, molding, drying, and firing, using SSA as a substitute for clay or some of the other additives (e.g., sand or sawdust). The present research [32,36–38] has shown justification for the production of bricks with added SS/SSA (maximum of 25 wt%), even though the results also differ between different authors indicating that results are influenced by the sludge origin, sludge treatment method, and the conditions used to generate the SSA.

## 2. Materials and Methods

### 2.1. Materials

SS samples for research purposes were provided by the WWTP Zagreb (ZG) and WWTP Karlovac (KA). The WWTP ZG uses secondary treatment and samples of fresh, digested, and mechanically dewatered SS were taken directly from in the outlet screw conveyor (September 2020). The WWTP KA uses tertiary treatment with phosphorus and nitrogen removal from wastewater which results in higher SS age and presence of coagulants in the SS. Samples of digested and mechanically dewatered SS were taken from a temporary landfill at which the SS was a few weeks old (October 2020). Dewatered SS was subjected to a drying process at 105 °C for 24 h, i.e., until a constant mass was obtained. Dried SS was then incinerated under laboratory conditions at a temperature of 900 °C. The resulting SSA material was odorless and brownish gray in color. SSA was then ground using a multi-purpose mill and sieved through a 700 μm sieve. The clay used in this study was provided by the company Termoterra, a brick manufacturer from Topusko (TO), Croatia (October 2020). The clay, like the SS, was also subjected to the same drying process. The dried clay was then ground with a mill and also sieved through a 700 μm sieve.

### 2.2. Characterization of SSA and Clay

The particle size distribution of clay and SSA was determined by the laser light scattering method, and the results are presented as plots of the percentage of particles of a given size in the particle volume (mass) distribution. The results of particle size distribution measurements made with a Malvern Mastersizer 2000 instrument are as follows: A representative sample was crushed, lightly pressed with a pestle in a porcelain mortar (note: all samples except clay could be crushed with fingers), then dispersed in water (approximately 100 mg in 1 mL) with an ultrasonic probe pretreatment (44 kHz,

190 W, steel tip, 3 × 20 s). Then the dispersed sample was added to the flow cell of the instrument and the particle size distribution was measured. A value characteristic of aluminosilicate, 1.65, was used as the reference refractive index.

The morphology of the samples was studied by scanning electron microscopy (SEM). The samples were examined at FE-SEM, Mira, Tescan, Czech Republic, and the evaporator was a Q150T, Quorum Technologies, England (the samples were evaporated with chromium for 100 s).

X-ray diffraction (XRD) analysis of the clay and SSA samples was performed. The powder samples were reduced by quantification to the amount required for analysis, homogenized, and placed in a carrier and subjected to X-ray diffraction (XRD) analysis on a Shimadzu XRD-6000 diffractometer using CuK α-radiation, with an accelerating voltage of 40 kV and a current of 30 mA, in the range of 2–90 2θ° with a step of 0.02 2θ° and a retention time of 0.6 s. Qualitative analysis was performed using the ICDD database of diffraction data and literature data. The results are shown graphically in (Figures 8–10) as a function of the intensity of the diffraction maxima around the diffraction angle.

Determination of oxides present in the clay and SSA was performed by atomic absorption spectroscopy using an AAnalyst 200 instrument (PerkinElmer, Inc., Waltham, MA, USA). Samples were first dissolved by boiling in an acid mixture in steel autoclaves with a Teflon cartridge, then diluted to the required amount and measured. Results are reported as mass percent of each oxide in the sample ($K_2O$, $MgO$, $Fe_2O_3$, $CaO$, $Na_2O$, $SiO_2$, $Al_2O_3$, and $TiO_2$).

Heavy metals in clay and SSA were determined by X-ray fluorescence (XRF), recorded with molybdenum tube and molybdenum secondary target, set at 1000 s at 45 kV/35 mA, under vacuum, and compared with IAEA SL-1.

### 2.3. Laboratory Production of Bricks and Testing of Their Properties

Some research has shown that initial laboratory-scale studies on smaller disc-shaped bricks may be useful in determining the optimum proportion of waste material in bricks [39–41]. Following the aforementioned work, solid disc-shaped bricks were produced in this research, diameter d = 50 mm, thickness s = 14 mm, weight m = 46–48 g. For the needs of brick production, two types of SSA were used—from WWTP Zagreb and from WWTP Karlovac—both incinerated at 900 °C (ZG 900 and KA 900). A total of seven types of bricks were produced—control bricks (without addition of SSA) made of 100% clay and bricks with SSA ZG 900 and SSA KA 900 in mass proportions of 5 wt%, 10 wt%, and 20 wt% (5 wt% ZG 900, 10 wt% ZG 900, 20 wt% ZG 900, 5 wt% KA 900, 10 wt% KA 900, and 20 wt% KA 900).

The mixing of the raw materials was done using an electric mixer (Figure 1. In this study, tap water was added to the solid raw materials. Bound water is the amount of water required to obtain a plastic moldable dough [42–44].

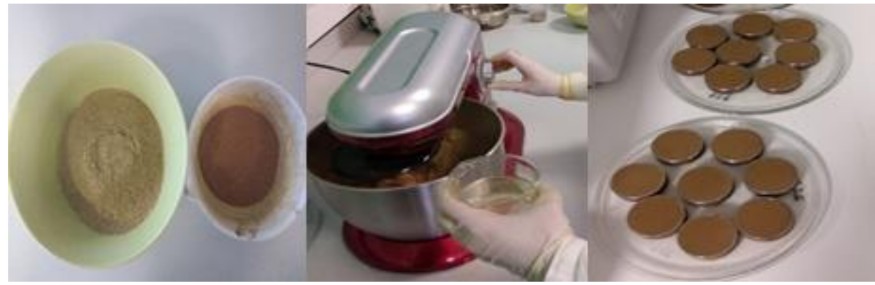

**Figure 1.** The brick manufacturing process in the laboratory.

A gradual increase in the SSA content in the mixture leads to a greater need for water addition due to the nature of non-plasticity of SSA (Figure 2).

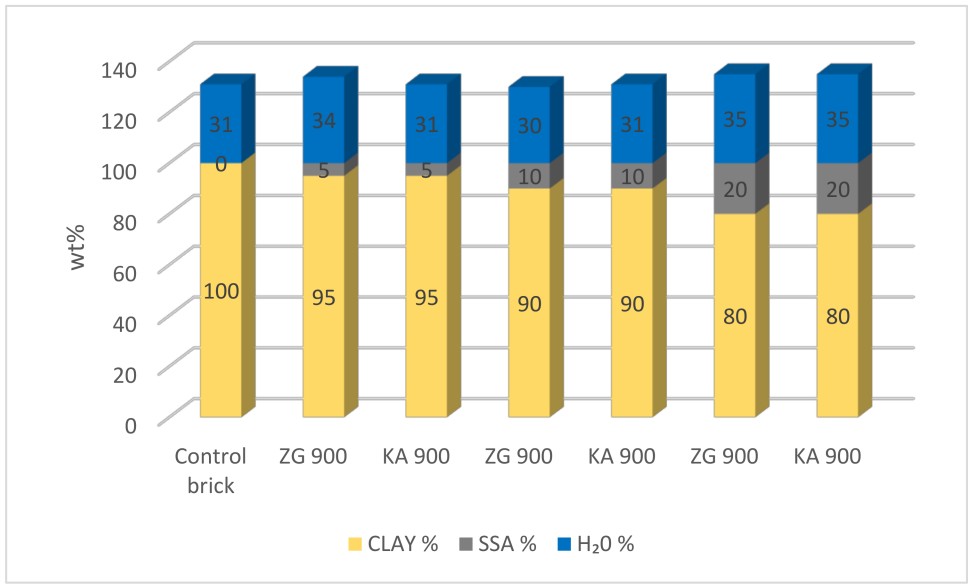

**Figure 2.** Water consumption in the manufacture of bricks with different proportions of SSA.

The prepared dough was then pressed into the metal molds by hand under load.

The bricks were left in the molds for a few hours (until the mass shrank), then it was released from the mold with light pressure, dried and stored until the next day (Figure 1). Drying was carried out in a forced-air oven at T = 105 °C, t = 24 h. After drying, the brick was left in the dryer until it had cooled down. The diameter of the brick was measured with a movable balance, and the mass of the brick was measured with an analytical balance.

The dry brick was placed in an annealing furnace and subjected to heat treatment with a temperature increase of 5 °C/min (Figure 3). After reaching a temperature of 950 °C, the bricks were fired for another three hours at the specified temperature and then left in the furnace until complete cooling.

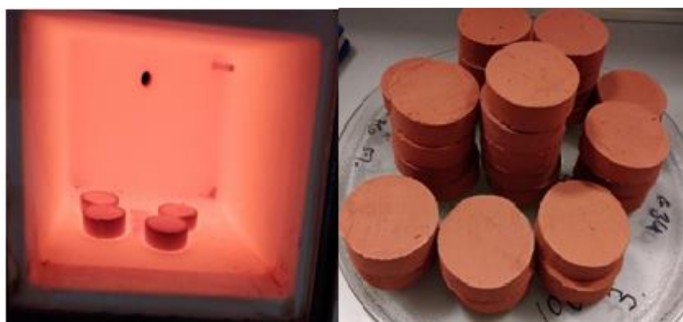

**Figure 3.** Firing process/fired brick.

The following tests were carried out on the bricks produced:

1.  Compressive strength

Sample preparation and testing was carried out in accordance with EN 772–1: 2015. The test is carried out on three specimens.

2.  Determination of water absorption of brick and sand–lime brick wall elements by cold water absorption.

The test was carried out in accordance with EN 772-21: 2011 on three specimens for each batch of bricks produced.

3. Determination of water absorption of brick wall elements with a layer for insulation against moisture by boiling in water. The test was performed in accordance with the standard EN 772-7: 2003 on three test specimens for each batch of bricks produced.

4. Saturation coefficient (water absorption by cold water/water absorption by boiling in water)

5. Determination of initial absorption by the method prescribed by ASTM C67/C67M—20, the test was carried out on three test specimens.

6. Determination of net volume and percentage of voids of brick wall units by hydrostatic weighing as per EN 772-3: 1998. The test was carried out on three specimens.

7. Loss during firing. The total mass loss during firing was determined from the expression

$$(m_{105°C} - m_{950°C})/m_{105°C} * 100\%. \tag{1}$$

## 3. Results

### 3.1. Characterization of Clay and SSA

#### 3.1.1. Particle Size Distribution

The particle size distribution or granulometric composition of clay plays an important role in the quality of brick products. Particles smaller than 2 µm are called clay fractions, those 2 to 20 µm are silt fractions, and particles larger than 20 µm are sand fractions [3,44]. The smallest fractions are also the richest in clay minerals and significantly affect the plasticity of brick clays. The increase in the mass fraction of particles below 2 µm contributes both to the increase in the mechanical properties of the product and to the increased shrinkage of the clay during the drying and firing process and the increase in the density of the product. When the mass fraction of non-clay particles is increased (e.g., when the sand content exceeds the allowable amount), the finished product is assumed to have a coarser texture and increased porosity [4,45,46]. By regulating the ratio of clay and non-clay minerals of the raw material, it is possible to model the plasticity properties of the raw brick [4,45–47]. By increasing the total amount of the largest fraction of the raw material, the shrinkage of the raw material is reduced, reducing the occurrence of cracks and deformation of the product. Increasing the total amount of clay minerals ensures not only the necessary plasticity, but also the strength of the raw product [45,48].

The particle size curves showed an approximately normal distribution for all three samples (Figure 4). A significant peak was estimated in the particle size range of 2.435–27.718 µm, with a volume ratio of 2.30–2.64. The range of particle size ratios was more than 70% and dominated by fine particles.

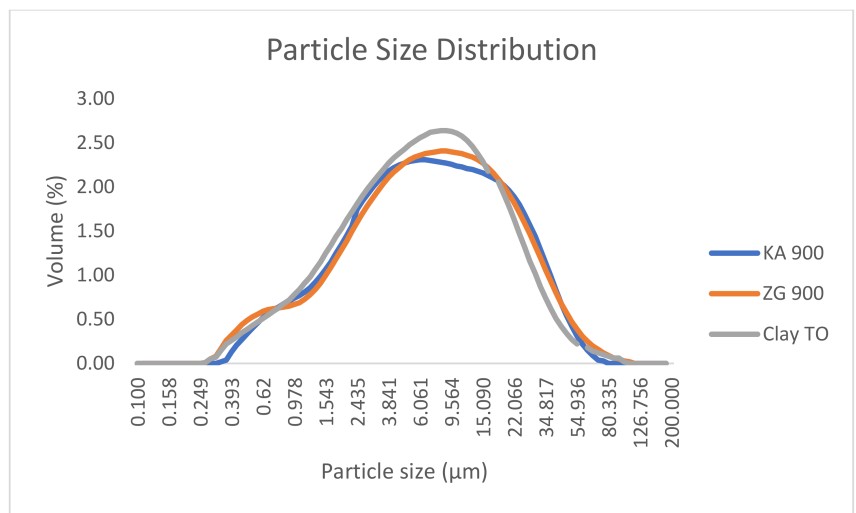

**Figure 4.** Particle size distribution Comparison of the particle size distribution of the clay sample with samples SSA ZG and SSA KA.

### 3.1.2. SEM

For both SSA samples (ZG 900 and KA 900), the microphotographs show a similar morphology to the clay sample (Figures 5–7). Polydisperse grains were observed. The particles were present in a wide range of sizes and with significant differences in shape. However, irregularly shaped particles predominated with a significant degree of agglomeration. Micrograms indicated a porous and smooth SSA structure. Glassy structures were also visible in both samples ($SiO_2$).

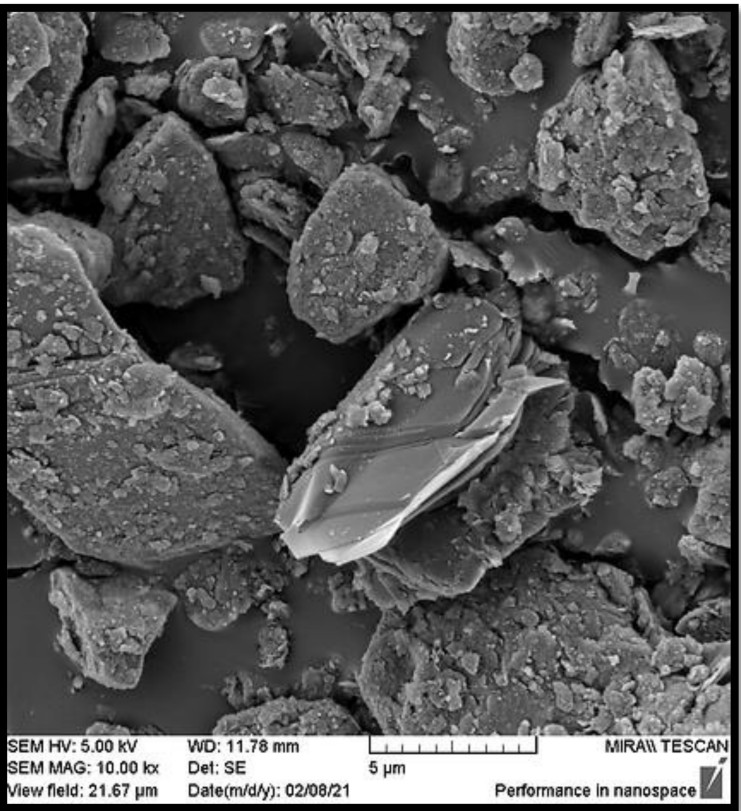

**Figure 5.** SEM microgram of clay.

Areas with different chemical composition were not well visible, indicating that the arrangement of crystals of different phases in the samples was homogeneous and that these crystals were very small. For this reason, it was not possible to observe significant differences in chemical composition in the morphology of samples

The SSA consisted of fine-grained, multiphase particles with many amorphous components. Because of the small grain sizes and transitions between grains of different compositions, there was also some "mixed" spectra represented by combinations of peaks of elements in adjacent grains.

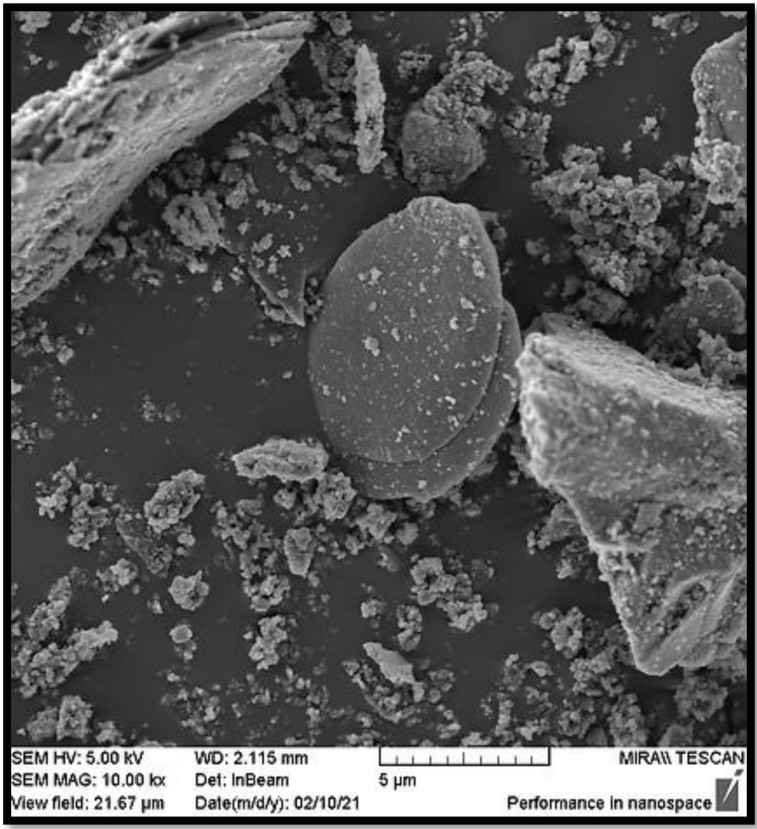

**Figure 6.** SEM microgram of SSA ZG 900.

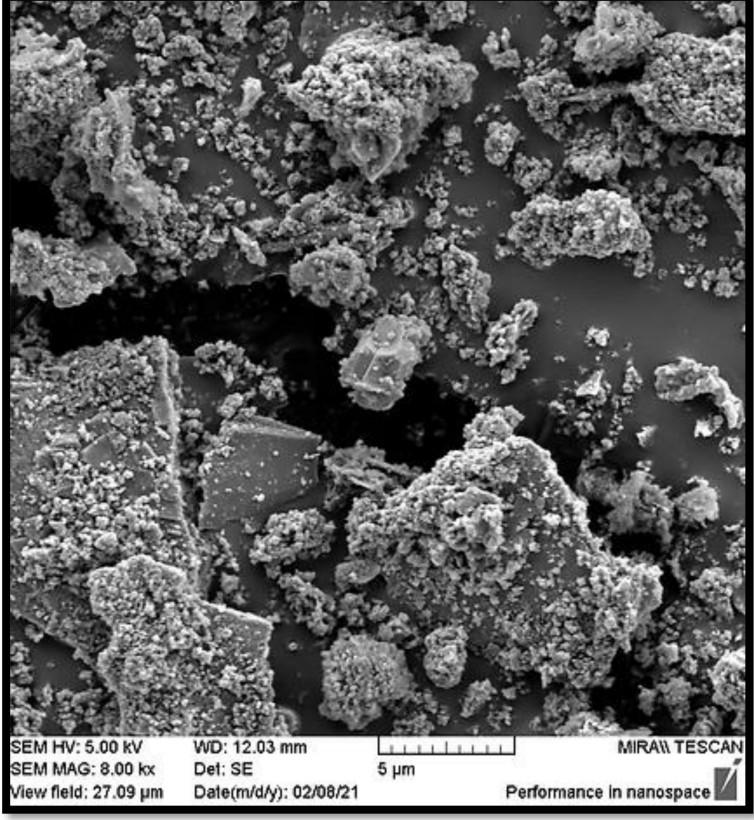

**Figure 7.** SEM microgram of SSA KA 900.

### 3.1.3. X-ray Diffraction Analysis (XRD) of Clay and SSA

The clay sample was based on clay raw materials. In the clay sample from the Topusko minefield, used in this research, the main phases were quartz, $SiO_2$ (ICDD PDF # 46-1045), and illite (ICDD PDF # 70-3754), and traces of low-grade cristobalite, $SiO_2$ (ICDD PDF # 76-0941) are found. The presence of stevensite (ICDD PDF # 25-1498) and birnesite (ICDD PDF # 43-1456) was possible in the traces. This cannot be confirmed but should be considered as indicative. The amorphous phase was also present in a very small amount in the clay sample (Figure 8).

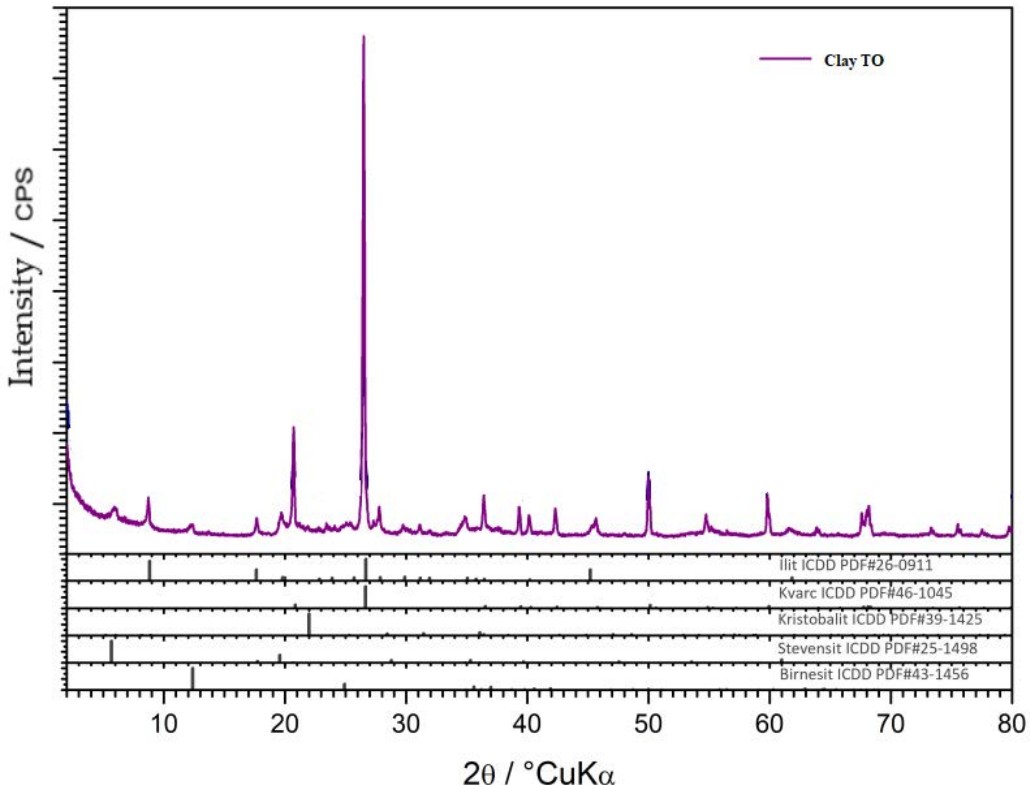

**Figure 8.** Clay mineral XRD analysis.

The SSA ZG 900 sample contained a visible amount of illite (ICDD PDF # 26-0911), quartz, $SiO_2$ (ICDD PDF # 46-1045), and cristobalite, $SiO_2$ (ICDD PDF # 39-1425). Aluminosilicate crystalline phases were present as secondary phases. An amorphous phase was not present (Figure 9).

The SSA KA 900 sample contained a visible amount of illite (ICDD PDF # 26-0911), quartz, $SiO_2$ (ICDD PDF # 46-1045), and cristobalite, $SiO_2$ (ICDD PDF # 39-1425). Aluminosilicate crystalline phases were present as secondary phases. An amorphous phase was not present (Figure 10).

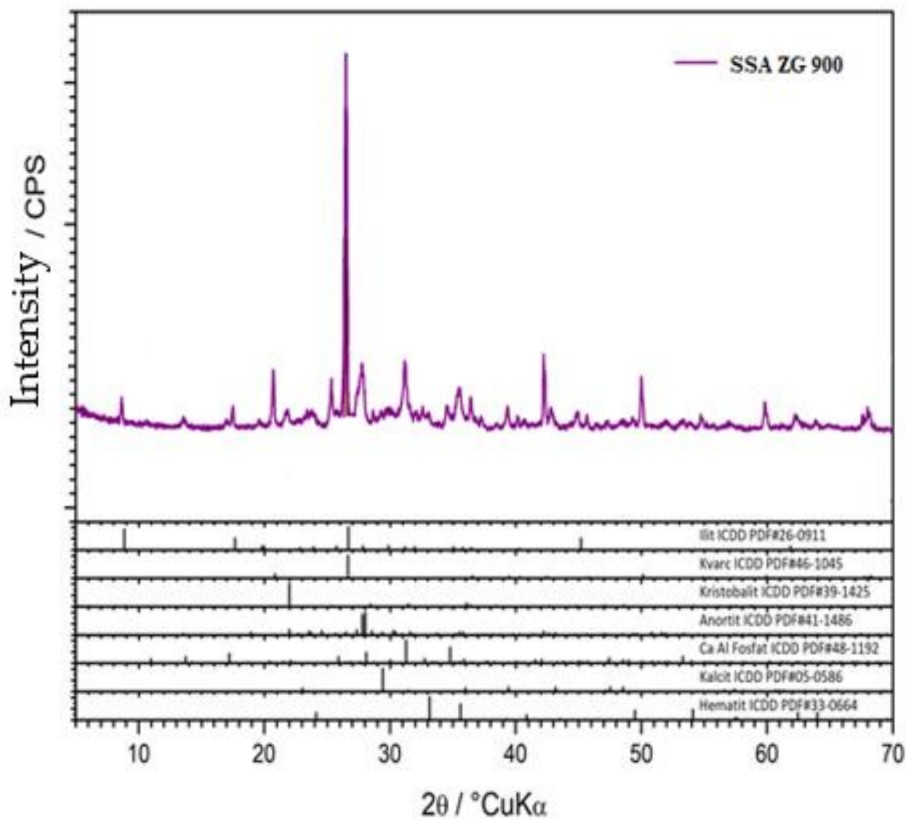

**Figure 9.** SSA ZG 900 mineral XRD analysis.

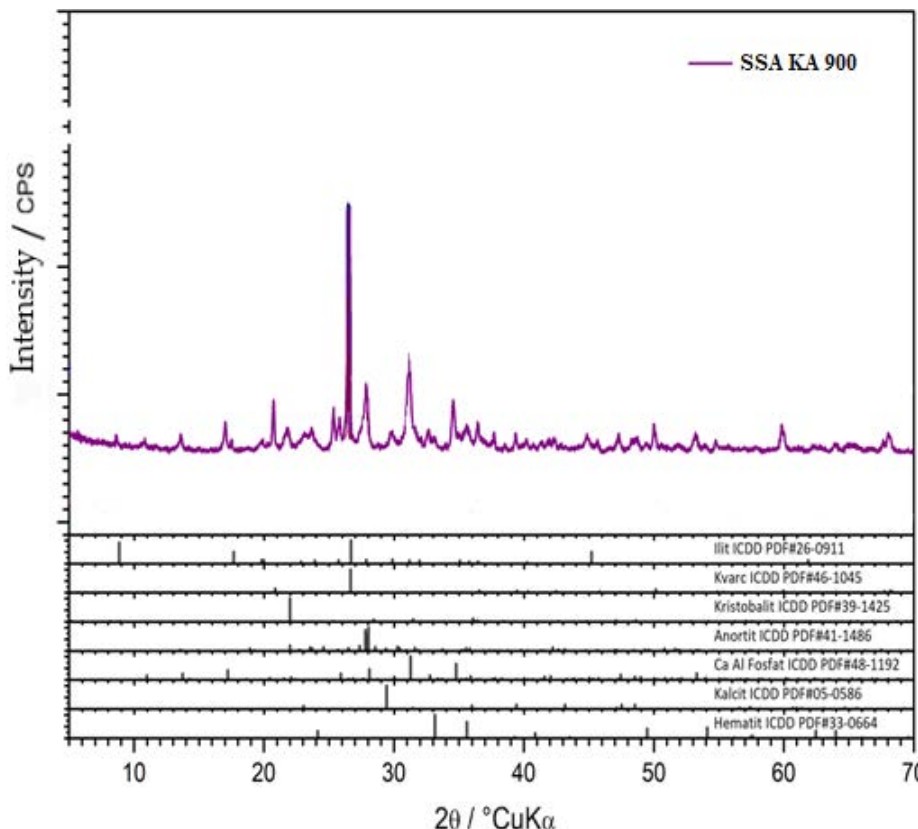

**Figure 10.** SSA KA 900 mineral XRD analysis.

### 3.1.4. Chemical Composition of Clay and SSA

Oxides

Based on a review of the research literature, it was found that the basic minerals that make up SSA are $SiO_2$ and $Al_2O_3$, which creates good conditions for use in the production of ceramic materials (building bricks, tiles, ceramic tiles, and glass ceramics) [49]. The content of CaO, $SO_3$, $P_2O_5$, and $Fe_2O_3$ is also significant. It was also found that these SSA usually contain high amounts of phosphate, typically 10–20 wt% in the form of $P_2O_5$, especially at the WWTPs with tertiary treatment.

The results show a high content of $SiO_2$ and $AlO_2$ in SSA, which are characteristic oxides for ceramic materials. In addition, SSA is characterized by a high content of CaO compared to clay (Figure 11). $Fe_2O_3$ is also a characteristic oxide in SSA and is normally 11.6% [49], but is slightly lower in the SSA ZG 900 and SSA KA 900. CaO and $Fe_2O_3$ contribute to better melting properties, which promote lower melting temperatures during the sintering process, thus reducing energy costs. However, the highest proportions of oxides are seen in sample of clay. Depending on the proportions of mineral components, clays may be non-carbonate (with low proportions of carbonate and higher proportions of quartz (over 50%)) or carbonate (with proportions of carbonate between 15% and 30% and a low proportion of quartz (less than 15%)) [3,4,50]. Non-carbonate clays contain a higher proportion of $SiO_2$ and $Al_2O_3$ in the chemical composition and carbonate clays contain a higher proportion of Ca and Mg [3,50]. Better resistance to wall elements is achieved by clays with lower CaO content [50–53].

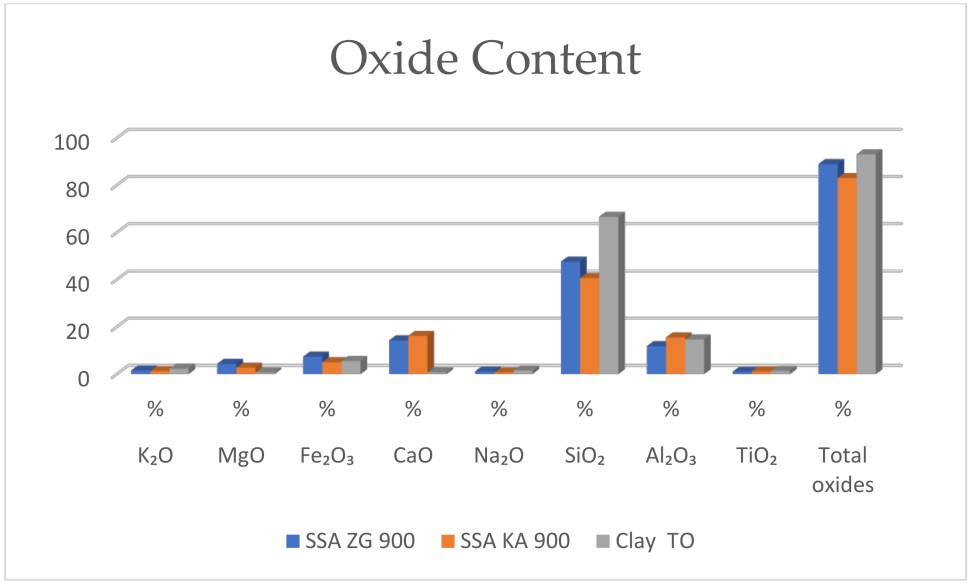

**Figure 11.** Properties of clay and SSA (mass fraction of each oxide in SSA ZG 900 and SSA KA 900).

Heavy Metals

The burning of organic material increases the concentration of heavy metals from the SS in the SSA by a factor of 2–3 [54] Elements in SSA that pose a potential threat are heavy metals [49]. Some heavy metals (Hg, Pb, and Cd) vaporize, but condensation reforms them on the surface of ash particles [55]. Significant vaporization of lead occurs at a temperature of ~400 °C, arsenic vaporizes at ~600 °C, while copper and zinc vaporize at ~800 °C [56]. Combustion dynamics affect the loss of particles and hence heavy metals. Lead and cadmium are largely contained in fine particles (1 μm); differences in chromium concentration may be related to particle size. Table 1 shows the results of the analysis of heavy metals in SSA ZG 900 and SSA KA 900 and compares them with the concentrations of heavy metals naturally occurring in clay. It can be seen that the origin of SS affects the concentrations of certain heavy metals in SSA. Heavy metals can dissolve even in low concentrations and

pollute the environment, indicating the need for safe disposal of the materials, which is confirmed by the analysis, as SSA is classified as a non-inert waste/residual [57]. Incorporation of SSA into bricks is an alternative method for stabilization of hazardous waste, which is characterized by reduction of leaching of hazardous substances (heavy metals) into the environment [49,58–61].

**Table 1.** Heavy metals in SSA and clay.

| Chemical Element | Measuring Unit | SSA ZG 900 | SSA KA 900 | CLAY (TO) |
|---|---|---|---|---|
| Fe | wt% | 3.61 | 2.15 | 4.15 |
| Zn | mg/kg DM | 931.26 | 2170.06 | 111.90 |
| Cu | mg/kg DM | 310.74 | 227.77 | 55.00 |
| Cr | mg/kg DM | 84.19 | 187.75 | 90.50 |
| Sr | mg/kg DM | 208.40 | 275.32 | 97.00 |
| Pb | mg/kg DM | 244.09 | 124.89 | 31.60 |
| Ni | mg/kg DM | 75.08 | 130.13 | 48.90 |
| V | mg/kg DM | 102.63 | 86.91 | 177.40 |
| As | mg/kg DM | 8.12 | <100 | 16.80 |
| K | wt% | 1.36 | 1.18 | 1.96 |
| Ca | wt% | 8.32 | 10.78 | 0.25 |
| Ti | mg/kg DM | 4160.12 | 3176.76 | 6721.00 |
| Mn | mg/kg DM | 950.46 | 461.91 | 313.00 |
| Ga | mg/kg DM | 13.09 | 13.70 | 22.80 |
| Br | mg/kg DM | 72.69 | 123.52 | <0.50 |
| Rb | mg/kg DM | 62.44 | 51.63 | 144.20 |
| Y | mg/kg DM | 51.69 | 41.85 | 106.90 |
| Zr | mg/kg DM | 341.49 | 368.88 | 632.00 |
| Th | mg/kg DM | 9.84 | 6.45 | 17.35 |

### 3.2. Properties of Bricks

Compressive strength is inversely proportional to the total porosity of the material, which means that bricks with low strength also have high porosity, making them susceptible to damage from the freeze–thaw cycle [3]. The minimum values of compressive strength of bricks proposed according to American and Canadian regulations are presented in Table 2 [62,63].

**Table 2.** Proposed minimum values recommended for clay bricks according to US and Canadian standards [62,64,65].

| | Standards | Proposed Minimum Value for Compressive Strength (N/mm²) | Proposed Maximum Value for Absorption at Boiling (%) | Proposed Maximum Value for the Coefficient of Saturation | Proposed Maximum Value for Absorption in Cold Water (%) |
|---|---|---|---|---|---|
| CSA | For one brick sample | 17.20 | 17.00 | 0.78 | 8.00 |
| | Mean value for five brick samples | 20.70 | - | - | - |
| ASTM | For one brick sample | 17.20 | 20.00 | 0.80 | - |
| | Mean value for five brick samples | 20.70 | 17.00 | 0.78 | - |

CSA, Canadian standards; ASTM, US standards.

The results show that the strength is strongly dependent on the amount of SSA in the brick. This finding is closely related to the amount of water absorbed as shown in Figures 12 and 13. It decreases with the increase of SSA content in the brick mixture. Bricks made of SSA ZG and SSA KA showed the best compressive strength at 5 wt%, while at 10 wt% compressive strength they were roughly equivalent to the control bricks. When the SSA content was increased up to 20 wt%, the compressive strengths showed significantly lower values compared to the other test specimens (Figure 12).

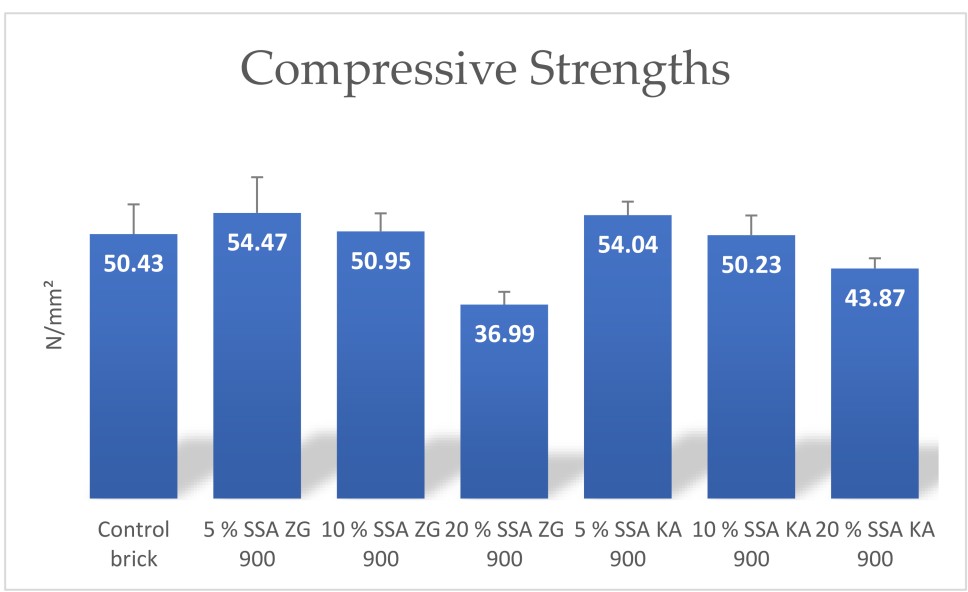

**Figure 12.** Mean values of compressive strengths of produced bricks containing SSA.

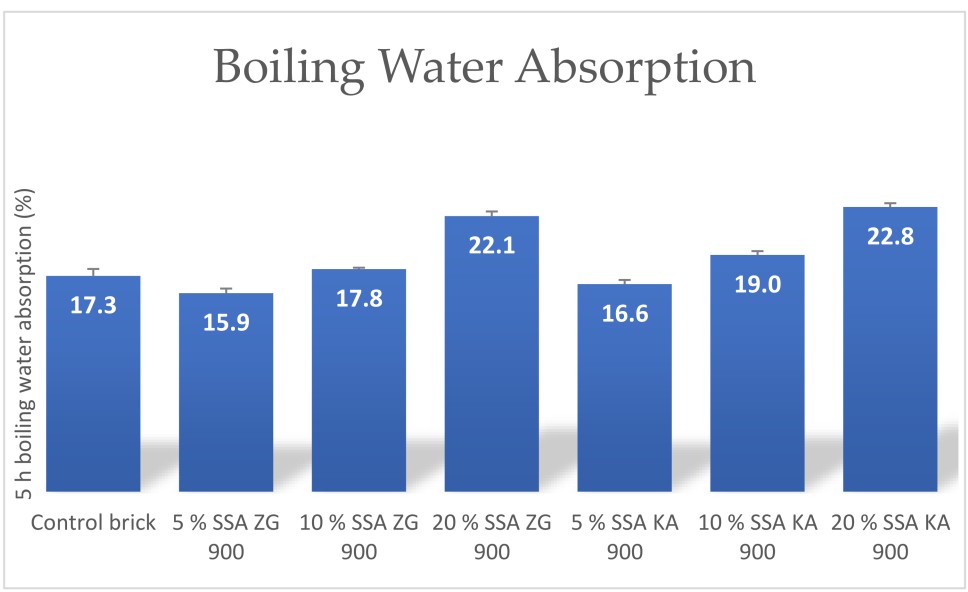

**Figure 13.** Average 5-hour boiling water absorption of bricks produced.

Essential parameters for the assessment of the freeze–thaw resistance of bricks are, in addition to the determination of the compressive strength, the water absorption during boiling, the water absorption in cold water, and the saturation coefficient.

Water absorption is an important parameter indicating the durability of the brick. Lower water absorption indicates greater resistance of the brick to external influences [62]. Replacing clay with SSA in higher mass proportions reduces the plasticity of the mixture, resulting in an increase in internal pores in the brick, which contributes to higher absorption [51]. Parameters of the water absorption during boiling and water absorption in cold water showed a linear growth with increasing proportion of SSA ZG and SSA KA in the brick, Figures 13 and 14. Water absorption should be between 5% and 20% for normally fired bricks [66].

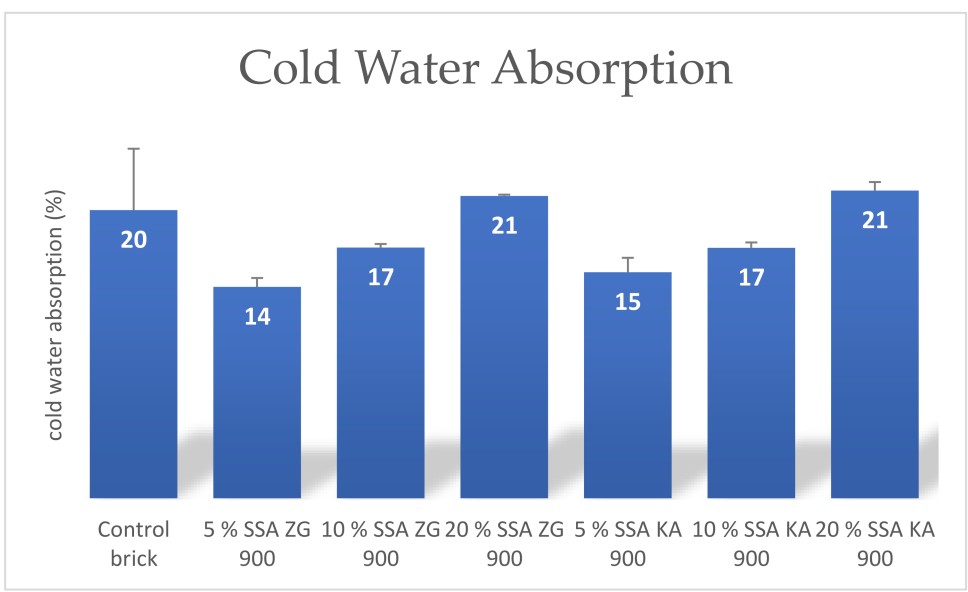

**Figure 14.** Average cold water absorption of the bricks produced.

The coefficient of saturation is the ratio between the absorption of cold water and the absorption of water by boiling. The saturation coefficient is used to determine the amount of water absorbed that will not damage the material by freezing/thawing. In other words, the saturation coefficient determines the pores that are easily filled with water in relation to the total pore volume [3]. Particle size distribution, i.e., a higher proportion of fine clay particles contributes to a higher saturation coefficient, while larger particles contribute to a decrease in the saturation coefficient [65]. The saturation coefficient was lower for all samples containing SSA compared to the control brick. For bricks containing 5%, 10%, and 20% SSA, the saturation coefficients were approximately the same (Figure 15).

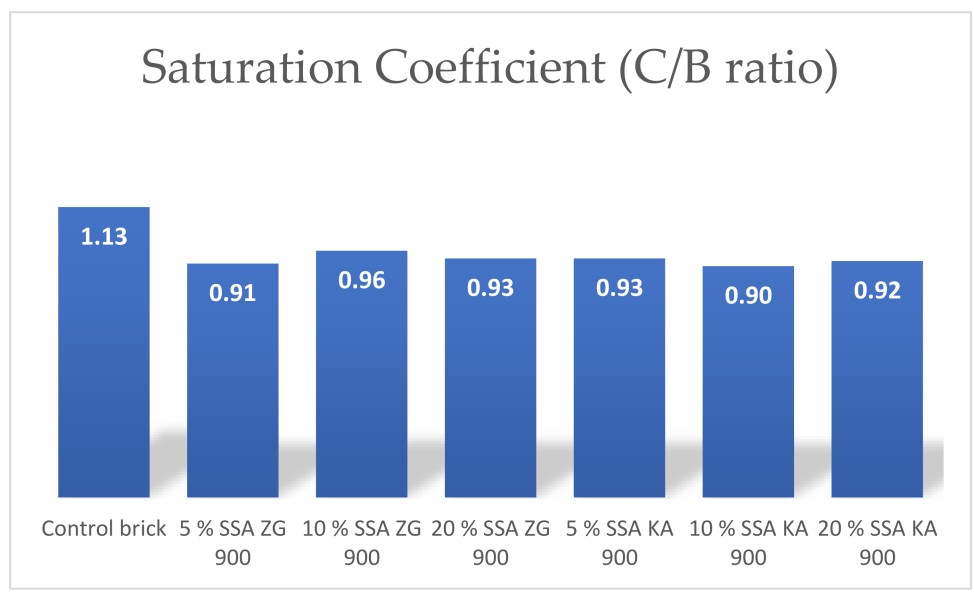

**Figure 15.** Mean values of the coefficient of saturation of the manufactured bricks.

The initial water absorption (IRA) is a measure of how quickly a brick will absorb water. The relationship between IRA and saturation coefficient was studied in [65] and a proportional increase in saturation coefficient with increasing initial water intake was found. The degree of correlation of these two parameters is a function of the homogeneity of the raw material and is more pronounced when only one raw material is used than

when multiple raw materials are mixed [4]. A higher rate of water absorption over time should indicate a higher proportion of large pores and thus greater resistance to freeze–thaw cycles [4,67]. This may be due to the texture of the sample surface. The initial water absorption parameter increased linearly with the increase of SSA content in the brick, and they were higher compared to the control brick (Figure 16).

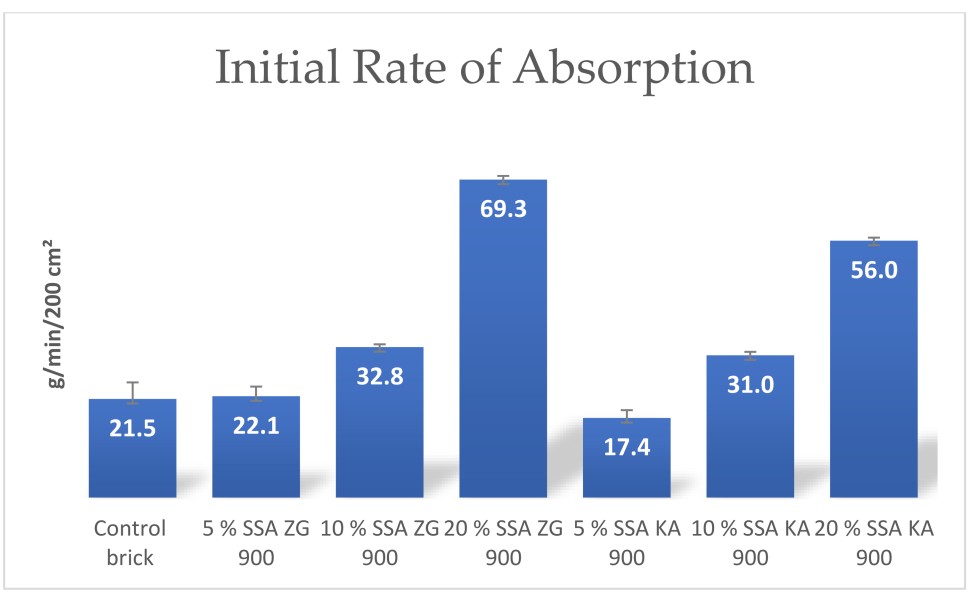

**Figure 16.** Mean values of initial adsorption rate of manufactured bricks.

The parameter percentage of voids in brick samples containing 5% and 10% SSA was significantly higher compared to control bricks, while bricks containing 20% SSA had a significantly lower percentage of voids (Figure 17).

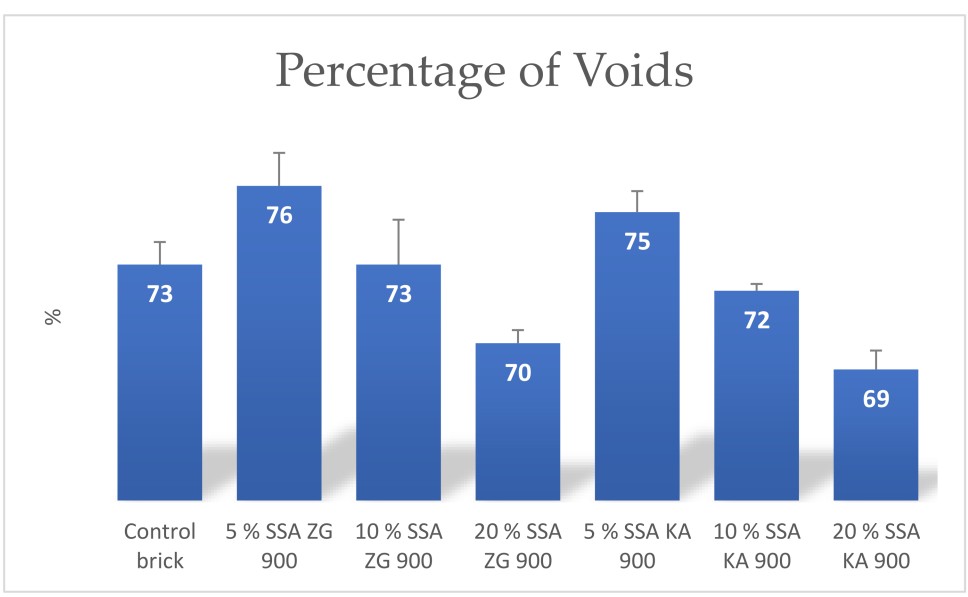

**Figure 17.** Mean values of the percentage of voids in the bricks produced.

Loss of mass during firing may be due to loss of chemically bound water and evaporation of compounds within the brick [62] and may be caused by combustion at high temperatures of both organic and inorganic substances present in SSA and clay [37,51]. The degradation of $CaCO_3$ is a major cause of weight loss during firing.

From the results presented, it can be seen that the SSA brick specimens showed a slightly higher weight loss during firing compared to the control brick, especially for the 10% content of SSA (Figure 18). The weight loss during firing was 15% for hand-molded bricks [58].

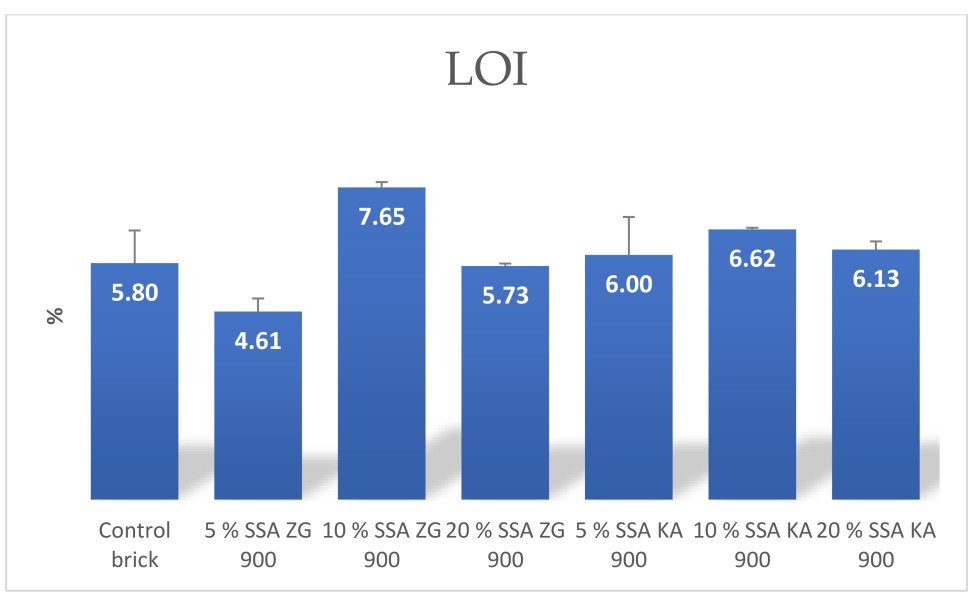

**Figure 18.** The mean values of the LOI of the manufactured bricks.

## 4. Conclusions

Studies on clay and SSA (ZG and KA) showed the compatibility of the two materials (grain size distribution, morphology, chemical composition, and mineralogical composition).

This study showed that replacing clay with SSA in proportions of 5 and 10 wt% resulted in profitable brick production with better or equal quality compared to control bricks. Increasing the SSA mass fraction to 20% showed a significant deterioration in brick quality.

Handmade bricks made of SSA showed only a slightly higher need for water addition at 20% SSA content compared to control bricks.

Compressive strength is the most important quality index for evaluating the performance of building materials. Partial replacement of clay with SSA in proportions of 5% and 10% resulted in bricks with high compressive strength. Bricks made with SSA content of 20 wt% showed significantly lower compressive strength compared to control bricks and bricks with lower SSA content.

The parameters of water absorption in cold and hot water recorded a linear increase with an increase in SSA addition.

The results of the initial absorption test on brick samples also recorded a linear increase with increasing SSA content in the brick and were higher compared to the control brick.

The saturation coefficients were approximately the same for all SSA proportions and slightly lower compared to the control brick.

The parameter percentage of voids in the brick recorded a linear decrease with increasing SSA content in the brick. The percentage of voids in the brick with 5% and 10% SSA was higher compared to the control brick, while the percentage of voids in the brick with 20% was lower compared to the control brick.

The LOI values of the tested bricks with different SSA contents showed slightly higher mass losses during firing compared to the control bricks.

The laboratory production and testing of the bricks give positive indications about the production of SSA bricks on a real scale and about the expected quality of the bricks obtained. From an economic point of view, the recycling of SSA in the brick industry is a good alternative and a positive example of a circular economy, given the ever-increasing

disposal prices and numerous restrictions on waste disposal. Calcium, iron, phosphorus, and potassium contained in SSA have fluxing properties that promote lower firing temperatures of bricks, resulting in energy savings.

**Author Contributions:** Conceptualization: N.S.; data curation: A.B., D.V. and K.N.; formal analysis: A.B. and D.V.; investigation: A.B. and D.V.; methodology: A.B., D.V. and N.S.; supervision: K.N.; validation: K.N.; writing—original draft, A.B. and D.V.; writing—review and editing: A.B., D.V. and N.S. All authors have read and agreed to the published version of the manuscript.

**Funding:** Croatian Science Foundation.

**Institutional Review Board Statement:** Not applicable.

**Informed Consent Statement:** Not applicable.

**Data Availability Statement:** Not applicable.

**Acknowledgments:** This work has been fully supported by the Croatian Science Foundation under the project "IP-2019-04-1169—Use of treated oily wastewater and sewage sludge in brick industry—production of innovative brick products in the scope of circular economy".

**Conflicts of Interest:** The authors declare no conflict of interest.

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
