# Peer review of "Use of Sewage Sludge Ash in the Production of Innovative Bricks—An Example of a Circular Economy"

_sustainability, doi:10.3390/su13169330_

Round 1

Reviewer 1 Report

Utilization of sewage sludge ash has high relevance, therefore the topic of manuscript sustainability-1305372 can be considered as interesting for the readers. Development of processes for enhanced utilization of sewage sludge and sewage sludge ash according to the main concept of circular economy can provide useful information for the science and for practice, respectively. The manuscript is generally well structured. The relevance of the study and research motivations is well defined. Applied methods (PSD, SEM, XRD, XRF, AAS, compressive test, water absorption/saturation, firing loss) are adequate to the investigated materials, products and the main aims of the experimental work. Materials and methods are given clearly.

Comments:

Please add the standard deviations for data presented in Fig. 9-16, and Table 1 (if available).

Figure 5 has poor quality (labels/text of the microphotos). I suggest the authors to improve the quality.

Text on Fig. 6-8 is not visible.

Some figure has frame, but others have not. I suggest the authors to unify the ’style’.

I suggest the authors to discuss briefly how affect the replacement of clay with SSA the economy of brick production and the price of the product.

Author Response

Dear Madam/Sir, 

I enclose corrections.

Best regards, 

Anđelina

Reviewer 2 Report

The paper represents the scientific level. The research methods are correct and properly described and the conclusions are resulted from research presented in the paper. However the discussion should be improved.  Please check punctuation one more time. The type of this paper should be added (article). Some specific comments are as follows:

  • Introduction: Please emphasize on works recently done- not older than 5 years. What is the hypothesis and objective of this paper?
  • Materials and Methods: All graphs should be prepared according to instruction for authors. The “wt%” for control brick is equal to 131%- what does it mean- Figure 1? L157-179. Please tabulate.
  • Results: All graphs should be prepared according to instruction for authors. L208. I don’t understand this statement (Figure 5…). Figures 5-8 are unreadable. Table 1 and 2- use dots (e.g. 3.61 not 3,61). Figure 13- what is on the Y-axis? The discussion should be improved. It is too poor. It also seems that in this paper are mainly figures...it doesn't look good (16 figures- in my opinion it is too many).
  • Conclusions: The conclusions are supported by the analysis of the data presented.

Author Response

(The authors gave the same response as above.)

Reviewer 3 Report

The manuscript can be very useful for the scientific community. However, I suggested to you many improvements crucial for achieving a high level. I am confident that you will be able to achieve my suggestions.

Lines 27-28: “Globally, 1.550 billion bricks are produced annually, 87% of which are produced in Asia”.

Please, provide some sources for such a sentence.

Lines 30-31: “(Indexbox, 2020)”.

You have to use numbers when you cite some sources. Furthermore, Indexbox is not available at the end of the manuscript among the references.

Lines 31-33: “Although the possibilities of using brick are wide, it is still mainly used for the construction of residential buildings”.

Please, provide some sources for such a sentence.

Lines 43-46: “In order to minimize the impacts associated with natural resource exploitation, reduce production costs, and establish a higher level of environmental sustainability, the nearly 50% of global manufacturers incorporate some form of waste into brick production.)”.

Please, provide some sources for such a sentence. In addition, you should remove the bracket at the end of the sentence.

Lines 49-52: “The possibility of sewage sludge (SS) has been studied since the 1980s [12] as a substitute raw material in brick production and the possibility of mixing SS with other waste materials (e.g. fly ash, ash from the bottom of the kiln, agricultural waste, forest waste, waste glass, etc.)”.

Thanks for the overview; it is very interesting. However, do you have any information about the energy saved by using sewage sludge ash in the production of bricks? It would be useful also to share such information.

Line 74: “indicatiog”.

Please, correct the typo writing “indicating”.

Section 2.1 (Materials).

Please, indicate when you got your samples.

Lines 85-86: “Dried SS were then incinerated under laboratory conditions at the temperature of 900 °C”.

In the abstract, you wrote “800 °C”. Furthermore, 900 °C is also mentioned in line 120. Please, check and clarify it.

Lines 99-100: “instrument instrument”.

Please, correct.

Line 124: “Torres et al. (2009), Guedes et al. (2014) and Ottosen et al. (2020)”.

Please, use numbers for citations.

Line 130: “control bricks”.

Please, provide detailed information concerning control bricks you considered.

Figure 4: “Clay TO”.

Please, clarify what you mean by “Clay TO”.

Line 206: “3.2.2 SEM”.

Please, clarify what you mean by “SEM”.

Line 226: “used in this research)”.

Please, correct the bracket.

Line 251: “Based on a review of the research literature”.

Please, provide some sources.

From line 275 (Heavy Metals).

You should also discuss the health risks related to using the three categories of bricks that you studied. Indeed, heavy metals concentration in bricks could affect people health. I suggest you to analyse and mention some studies about it. It is something that should not be underestimated. Maybe you could even add a further section about this topic. If few studies exist, you should highlight it, suggesting future research on it.

Line 301: “Straube et al., 2010”.

Please, use numbers for citations.

Table 2 (line 304): “US and Canadian standards [53]”.

The reference you mentioned [53] is very old (i.e. 1993). You should consider more recent standards and related references.

Conclusions (from line 371).

You should also discuss how to conduct future research. Furthermore, you should highlight how your research can already be beneficial for the scientific community.

Finally, I would also suggest you consider the health aspects related to the concentration of some contaminants that can be found in control and sewage sludge ash bricks (you already showed it in Table 1, but you did not discuss the health aspects). It is something rarely discussed, but it can be very instrumental.

Author Response

(The authors gave the same response as above.)

Reviewer 4 Report

Interesting paper about the use of sewage sludge ash in bricks.

In general, the manuscript is well presented but some information should be clarified or completed:

  1. Figure 4: on the YY axis, the variable recorded is “volume” or “weight” (in %)?
  2. Figure 5: can you include a legible “scale bar” in these SEM images?
  3. Figures 6 to 8: you can also include the name of the elements in each main peak;
  4. Line 263: what does “… samples of raw Clay TO.” means?
  5. Nothing is written about the smell of these sustainable bricks. The bad(?) smells of the SSA was completely eliminated in the oven (@950 °C)?
  6. Can the Authors include any more information on the estimated availability (in each wastewater treatment plant) and costs of processing this sludge?
  7. If appropriate, the Authors can also highlight some information regarding other important question: contamination of soils (with heavy metals, etc.) will be a problem? (from leaching, when bricks are exposed to real environmental conditions);
  8. Another issue that can be addressed: the expected quality for the adherence between the surface of these bricks and the cement mortar or paints;
  9. Section 5 (Conclusions): some conclusions can be better summarized;
  10. Only ~6 references (and 3 of them are internet sites) are less than 5 years old. Can you include some other more recent references?

Other minor remarks:

  1. In “Abstract”: please change commas (,) by points (.) in the decimal separator of the numbers;
  2. The same in Tables 1 and 2;
  3. Line 74: change “indicatiog” by “indicating”;
  4. Line 80: remove one “the” in “… from the the outlet screw …”;
  5. Line 81: change “phosporus” by “phosphorus”;
  6. Line 82: change “presense” by “presence”.

Author Response

(The authors gave the same response as above.)

Round 2

Reviewer 1 Report

Manuscript deals with an interesting topic that has practical relevance, as well. Authors have revised the manuscript thoroughly according to reviewers' comments and suggestions-. The overall scientific quality of manuscript has improved significantly due to the revision. 

Author Response

(The authors gave the same response as above.)

Reviewer 2 Report

Please pay attention on graphs. All graphs should be clear for readers.

Author Response

(The authors gave the same response as above.)

Reviewer 3 Report

I am glad you followed most of my suggestions.

However, please, recheck the following oversight concerning line 319 (Table 1. Minimum values recommended for clay bricks according to US and Canadian standards [62, 63]): 

  • This is Table 2.
  • I think you probably mentioned the wrong references (indeed, Trauner, which you cited in the first version of the manuscript, is now [64]). Is it correct?

Author Response

(The authors gave the same response as above.)
